# Assessing COVID-19 Effects on Inflation, Unemployment, and GDP in Africa: What Do the Data Show via GIS and Spatial Statistics?

**Butte Gotu and Habte Tadesse \***

Department of Statistics, Addis Ababa University, Addis Ababa P.O. Box 1176, Ethiopia
\* Correspondence: habte.tade@yahoo.com

**Abstract:** What are the effects of Corona Virus Disease 19 (COVID-19) on inflation, unemployment, and GDP in Africa? Using geo-coded cross-sectional data taken from the World Health Organization and International Monetary Fund, we investigate the spatial distribution of COVID-19 and its effects on inflation, unemployment, and Gross Domestic Product (GDP) in Africa by employing the Geographic Information System (GIS), multivariate analysis of covariance (MANCOVA), and spatial statistics. The entire dataset was analyzed using Stata, ArcGIS, and R software. The result shows (1) that there is evidence of a spatial pattern of COVID-19 cases and death rate clustering behavior in Africa, verifying the existence of spatial autocorrelation. The result also reveals (2) that COVID-19 has a negative effect on unemployment, inflation, and GDP in Africa. We confirmed that (3) temperature, rainfall, and humidity were statistically significantly associated with the spread of the COVID-19 pandemic in Africa. The comparison of the GDP of African countries before and after the pandemic shows (4) a large decrease in GDP, the highest in Seychelles (23 percent). The result of the study shows (5) that there has been a significant increase in inflation and unemployment rates in all countries since the outbreak of the pandemic as compared to the time before the outbreak. There is also evidence that (6) there is a significant relationship between death rate due to COVID-19 and population density; temperature with COVID-19 cases and death rate; and precipitation with death rate due to COVID-19. Therefore, respective governments and the international community need to pay attention to controlling/reducing the impact of COVID-19 on inflation, unemployment, and GDP, focusing on the indicated demographic and environmental variables.

**Keywords:** spatial pattern; GIS; GDP; COVID-19; inflation and unemployment

## 1. Introduction

*Study Background*

In late 2019, a new virus called Corona virus was seen in China and spread alarmingly to different parts of the world [1]. It was declared a novel virus, COVID-19, and characterized as a pandemic in early 2020 [2].

The global economy has been largely affected by the spread of the pandemic. Thus, clearly examining its relationship with different factors has been the interest of many researchers. It was well noted that temperature [3,4], humidity [5], sunshine hour [6], precipitation [7], wind condition [8], air quality [9,10], and population density [11,12] are regarded as the most driving factors for the spread of the pandemic. The 2021 World Economic Outlook from the International Monetary Fund (IMF) shows that global economic growth fell to an annualized rate of around −3.2% in 2020, with a recovery rate of 5.9% projected for 2021 and 4.9% for 2022.

The report by [13] revealed that in Sub-Saharan African countries, the global shock was projected to contract economic activity by 2.8% in 2020 from 2.2% in 2019. Similarly, it was noted that the massive spread of the pandemic varied from continent to continent [14–16].

Moreover, the articles by [17] assessed the impacts of covariates on COVID-19, while [18] addressed the prediction of COVID-19 in Africa; however, they lacked an impact assessment related to the economy. According to a report by the World Bank in 2017, Sub-Saharan Africa paid 5.17% of its total GDP for health, which is smaller than the 9.89% contributed by the Organization for Economic Co-operation and Development. This indicates that the amount of money contributed to the health care system was not enough to address the pandemic [19].

The spatial distribution of COVID-19 varies across the globe, including in Africa. On February 14, the first case of COVID-19 was reported in Africa, while the first South African case was confirmed on March 5, 2020 [20]. Following this, the pandemic has spread rapidly across the entire African continent. To mitigate its spread by paying attention to the hotspot areas, it is believed that the Geographic Information System (GIS) provides an excellent medium for integrating specific health data and economic data, along with its understanding of population habits, including healthcare services, and the environment [21]. Understanding disease space and time dynamics is important for both planners and epidemiologists, as with space distribution, the hot spot areas are marked for intervention [22]. Thus, clearly determining the spatial distribution of the pandemic and its impact, along with employment and inflation on the GDP, is important for hotspot area identification and further intervention.

A considerable number of articles have been published based on GIS and spatial modeling since the outbreak of the pandemic. For instance, [23] presented how various GIS applications are taken into consideration to model the driving factors of the pandemic, but it gives less emphasis to its impact on the economy. Spatial models are the most widely used tools to link the geographic relationship between several predictor parameters associated with the outbreak of any pandemic [24,25]. Moreover, GIS and spatial autoregressive models were the most popular techniques to describe the dispersion of COVID-19 by incorporating the issues of spatial dependency while considering population density, level of healthcare services, and environmental variables [26].

This work focuses on the spatial distribution of COVID-19 and tries to identify the effect of COVID-19 on inflation, unemployment, and GDP in Africa, along with the driving forces of the pandemic. It also examines the effect of the pandemic on three dominant economic indicators, namely inflation, unemployment rate, and GDP per capita, and suggests ways of reducing the impact of the pandemic and actions to be taken for further intervention. In short, this paper is aimed at assessing COVID-19 and examining its effect on inflation, unemployment rate, and Gross Domestic Product (GDP) in Africa via GIS and spatial exploratory techniques.

## 2. Data Source and Methodology

### 2.1. Study Area and Period

Africa is regarded as the second-largest and most populous continent after Asia. The continent has a land area of about 30.3 million km$^2$, including adjacent islands; it covers 6% of Earth's total surface area and 20% of its land area. Africa's population was about 1.3 billion in 2018, which accounts for about 16% of the world's human population. The size of the population is estimated to be 1.4 billion in 2022 (16.7% of the world's population). The map of the study area is presented in Figure 1. A total of 54 African countries were considered for the study.

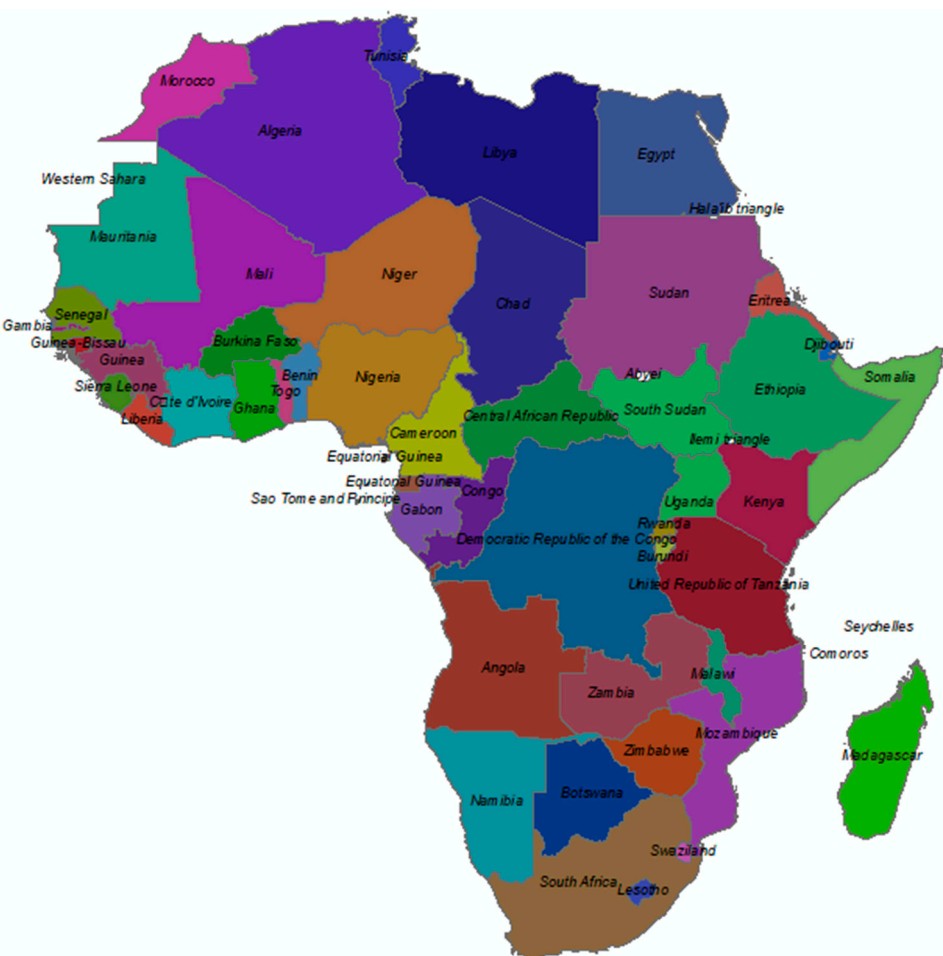

**Figure 1.** The map of 54 African countries.

*2.2. The Data*

Invoked with the initiation of the current impact of cross-cutting issues, such as the outbreak of the pandemic, war, and inflation, including the unemployment rate, global datasets are taken into consideration. Four global datasets were considered: COVID-19 cases and deaths from the World Health Organization: GDP per capita from the International Monetary Fund; inflation taken from the different global databases; and unemployment rate directly taken from the World Bank, referring to the unemployment data that is based on the total labor force in line with the proposed objectives. The entire dataset that inspired this study was secondary data collected directly from the four datasets indicated above. The data were compiled from the World Health Organization's reports up to 1 January 2021, since the outbreak of the pandemic. This data set contains the number of COVID-19 cases, deaths, and recoveries for all African countries. Moreover, the data for climate and other parameters were collected from online meteorology. The data on population density and GDP were taken from Richest African Countries 2022 (World Population Review, 2022). ArcGIS 10.4.1 was used for mapping the spatial distribution of COVID-19 and GDP per capita.

*2.3. Variable Identification*

The outcome variable in the study is the gross domestic product (GDP), measured in billions of US dollars.

The predictor variables considered in this study are summarized in Table 1.

**Table 1.** Summary of the predictor variables.

| Category | Variables | Descriptions |
|---|---|---|
| Meteorological factors | Temperature | The yearly average temperature in degrees Celsius |
| | Relative humidity | The daily average relative humidity in percentage |
| | Precipitation | Daily average wind speed in km/hr |
| | Population size | Population density for each country in Africa |
| Inflation | Consumer price index | Global database |
| Unemployment rate | Unemployment rate based on the total labor force | Global database (unemployment data and total labor force) |

*2.4. Spatial Statistical Analysis*

In this study, the methods of statistical analysis used include descriptive statistics, spatial autocorrelation analysis, and a spatial autoregressive model. Following the analysis, the model adequacy checks (diagnostics) for fitted models were examined.

2.4.1. Concept of Spatial Autocorrelation/Dependence

The primary premise underlying the analysis of spatial data is that values of a variable in close proximity are more similar or related than values in distant locations.

Tobler's first law of geography summarizes this inverse relationship between value association and distance: "Everything is related to everything else, but near things are more related than distant things" [27]. When close-by observations (i.e., those in the same place) have similar variable values, the pattern is said to have positive spatial autocorrelation (self-correlation). In contrast to Tobler's law, negative spatial autocorrelation is stated to exist when observations that are close in space are more dissimilar in variable values than observations that are further apart. When variable values are independent of location, zero autocorrelation exists.

According to [27], spatial autocorrelation can be loosely defined as the coincidence of value similarity with location similarity. In other words, high or low values for a random variable tend to cluster in space (positive spatial autocorrelation), or locations tend to be surrounded by neighbors with very dissimilar values (negative spatial autocorrelation). Of the two types of spatial autocorrelation, researchers usually focus on positive autocorrelation. Negative spatial autocorrelation implies a checkerboard pattern of values and does not always have a meaningful substantive interpretation [28]. The spatial autocorrelation/dependence is the situation where the dependent variable (the error terms) at each location is correlated to the observation in the dependent variable (the error term) of the other location [29–32].

2.4.2. Methods of Measuring Spatial Autocorrelation
Contiguity Spatial Weight Matrix

According to Tobler's first law, the neighboring or nearest regions are coded in the form of a spatial weight matrix with zero diagonal and non-zero off-diagonal elements, which often weigh to sum to unity in each row with typical elements. The non-standardized weight matrix is stated as:

$$w_{ij} = \begin{cases} 1, & \text{if } i \text{ is neighbor to } j \\ o, & \text{other wise} \end{cases} \tag{1}$$

The standardized weight matrix, which incorporates the average weight value, can be written as follows:

$$W_{ij} = \frac{w_{ij}}{\sum_{j=1}^{n} w_{ij}} \tag{2}$$

where $w_{ij}$ are elements of the non-standardized weight matrix along location $i$ and location $j$ and $\sum_{j=1}^{n} w_{ij}$ is the $i$th row total of the non-standardized weight matrix; $n$ is the number of locations considered.

### 2.4.3. Test of Global and Local Spatial Autocorrelation

Spatial autocorrelation is an important concept in spatial statistics, and it is used to measure similarity between nearby observations. The test for spatial autocorrelation is designed to quantify the extent of clustering and allow for statistical inference. The null hypothesis (under the normality and independence assumptions) is given by:

**H$_0$.** *There is no spatial autocorrelation ($\rho = 0$)*

Against the alternative hypothesis of spatial dependence/autocorrelation ($H_1 : \rho \neq 0$) which is the claim of interest. To test this hypothesis, we have used Moran's I and Geary's C analyses.

### Moran's I Correlation Analysis

Moran's I correlation coefficient is widely used to identify the spatial association pattern. To compute the spatial autocorrelation of the death rate and confirmed cases due to COVID-19, Moran's I correlation was employed. The values of Global Mora's I lie in the interval of $-1$ and 1. If the value is significantly less than 0, then there is a negative spatial relationship; if it is greater than zero, there is a positive spatial relationship; and if it is zero, there is no spatial relationship. Local Moran's I was used to identify the local spatial pattern and outliers among the death rate and confirmed cases due to COVID-19 in all African countries. This analysis was conducted using ArcGIS version 10.4 and GeoDa. The global Moran's I is given as follow:

$$I_g = \frac{n}{\sum_{i=1}^{n} \sum_{j=1}^{n} w_{ij}} \frac{\sum_{i=1}^{n} \sum_{j=1}^{n} W_{ij}(y_i - \overline{y})(y_j - \overline{y})}{\sum_{i=1}^{n} (y_i - \overline{y})^2} \tag{3}$$

where $I_g$ is global Moran's I correlation coefficient; $y_i$ and $y_j$ respectively represent either the death rate or confirmed cases due to COVID-19 of site $i$ and site $j$; $\overline{y}$ is the mean of $y$; $n$ is the number of locations (countries); and $W_{ij}$ is the spatial weights between site $i$ and site $j$.

### Moran Scatter Plot

The Moran scatter plot is a useful visual tool that enables us to assess how similar an observed value is to its neighboring observations. It is used to show the linear correlation between the dependent (outcome) variable (Y) and the corresponding neighboring dependent variable (WY). Specifically, WY is plotted against Y, and the Moran's I coefficient is the slope of the regression curve [27].

The four different quadrants of the scatter plot correspond to the four types of local spatial association between a region (country) and its neighbors: the first quadrant (HH) is a region with a high value surrounded by regions with high values (top on the right); the second (LH) is a region with a low value surrounded by regions with high values (top on the left); the third (LL) is a region with a low value surrounded by regions with low values (bottom on the left); and the last (HL) is a region with a high value surrounded by regions with low values (bottom on the right). The first and third quadrants refer to positive spatial autocorrelation, indicating spatial clustering of similar values, whereas the second and fourth quadrants represent negative spatial autocorrelation, indicating spatial clustering of dissimilar values.

### 2.5. Spatial Statistical Methods of Analysis

Spatial exploratory analysis, mainly the Moran scatter plot test of spatial autocorrelation by incorporating the spatial weight matrix, was employed to examine the issue of spatial autocorrelation [28,33–37]. The model used to describe the outcome variable

is described in Section 2.3. The effect of the climatic factors on COVID-19 was tested with the help of the Multivariate Analysis of Covariance (MANCOVA). Finally, the spatial autoregressive model is taken into consideration.

## 3. Results and Discussion

### 3.1. Descriptive Results

The African economy had been progressing prior to the outbreak of the pandemic. However, according to the recent report by the African Economic Outlook (2022), about 30 million people in Africa were pushed into extreme poverty in 2021, and about 22 million jobs were lost in the same year because of the pandemic. It was projected that the trend would continue, for various reasons, through the second half of 2022 and on into 2023. Thus, assessing COVID-19, inflation, and unemployment and their impacts on Gross Domestic Product (GDP) in Africa is highly recommended and advisable. This section mainly focuses on descriptive results and discussions based on the global datasets.

Table 2 presents a summary of descriptive measures on gross domestic product in billions of dollars, death rate per 1000 people, and COVID-19 cases for all 54 African countries. The data were obtained from the WHO and the International Monetary Fund.

**Table 2.** Summary of descriptive measures.

| Variables | Number of Countries | Mean |
|:---:|:---:|:---:|
| GDP per capita (USD) | 54 | 62.78 |
| Deaths per 1000 people | 54 | 4.154 |
| COVID-19 cases (per 1000 people) | 54 | 206.013 |

### 3.2. The Top Seven African Countries Affected by COVID-19

Based on the data, it was noted that Angola, Algeria, Egypt, South Africa, Kenya, Tunisia, and Seychelles were regarded as the top seven countries with a high number of COVID-19 cases. Among these, South Africa is the country with the highest number of COVID-19 cases (Figure 2).

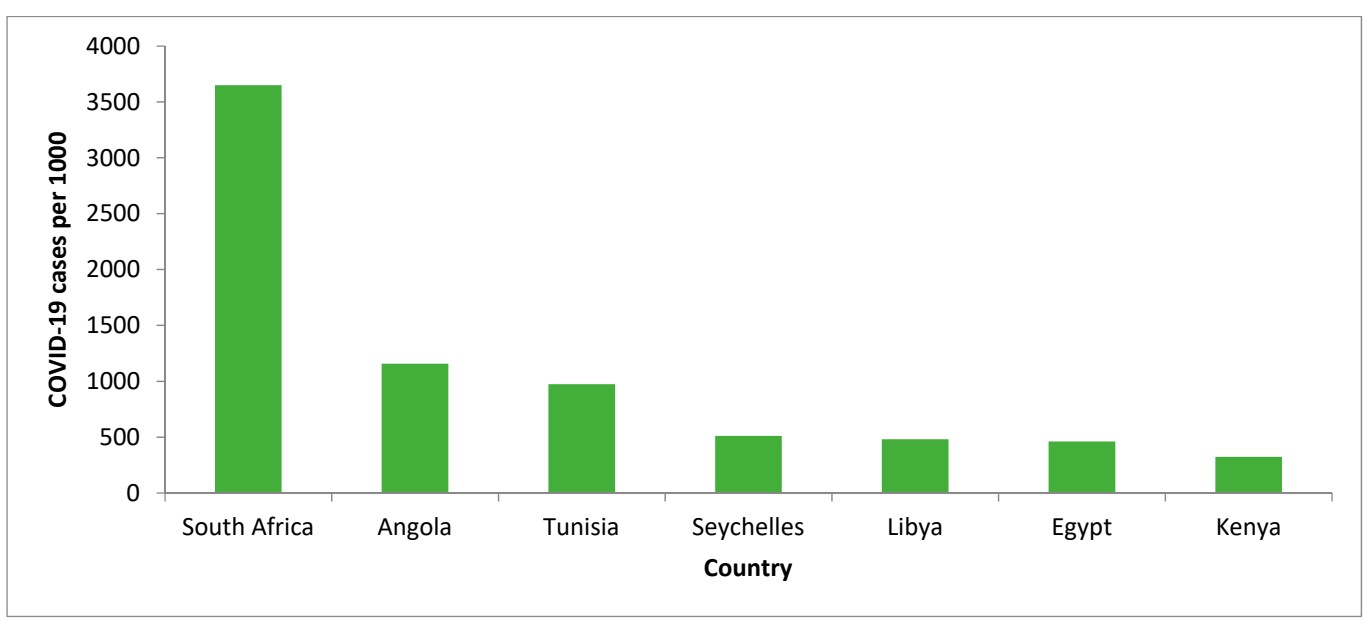

**Figure 2.** Hotspot has confirmed cases of COVID-19 in Africa.

To examine the effect of COVID-19 on GDP per capita, the top seven African countries with high COVID-19 cases and deaths were taken into consideration. Figure 3 indicates

the magnitude of GDP per capita before the pandemic (2019) and during the pandemic (2021), where the *x* axis denotes the name of the countries (Angola, Algeria, Egypt, Kenya, Tunisia, South Africa, and Seychelles), and the *y* axis indicates the GDP per capita in US dollars. It can be seen that the GDP per capita has decreased during the pandemic as compared to the GDP per capita before the outbreak of the pandemic in all countries except Egypt. The increase in Egypt's GDP is largely accounted for by the fact that during the pandemic, a large amount of money was allocated to control and prevent COVID-19 and the government's swift and prudent policy response, coupled with significant IMF support. Moreover, the country has experienced fewer movement restrictions and implemented a Preparedness and Response Plan (CPRP) (United Nations Egypt, 2020). The result based on the mean difference of the paired sample t test confirms a significant mean difference in GDP before and after the outbreak of the pandemic (Figure 3).

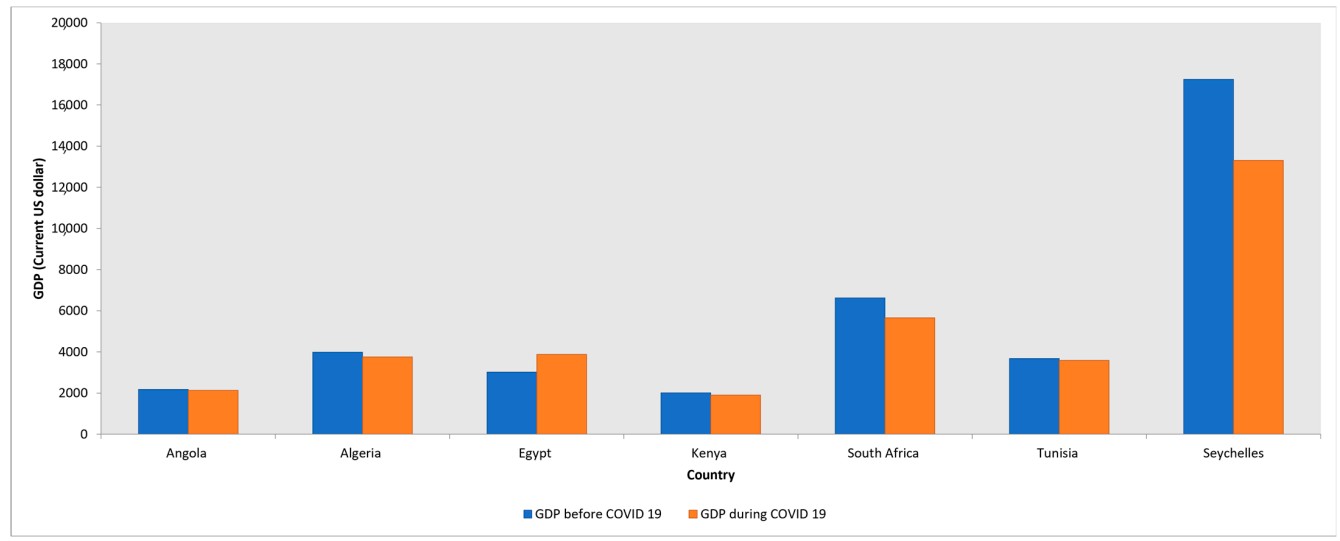

**Figure 3.** GDP per capita before and after the outbreak of COVID-19 in seven African countries.

As indicated in Table 3, the GDP per capita of Angola decreased from 2177.8 to 2137.9 (1.8%) dollars, Algeria from 3989.7 to 3765.0 (5.6%) dollars, Kenya's GDP per capita decreased from 2006.8 to 1909.3 (4.9%) dollars, and South Africa's GDP per capita decreased from 6624.8 to 5655.9 (14.6%) dollars.

**Table 3.** Summary of GDP per capita before and during COVID-19.

| Country | GDP in USD (before COVID-19: 2019) | GDP in USD (during COVID-19: 2021) | Percentage Decrease |
|---|---|---|---|
| Angola | 2177.8 | 2137.9 | −1.8 |
| Algeria | 3989.7 | 3765.0 | −5.6 |
| Egypt | 3019.1 | 3876.4 | 28.4 |
| Kenya | 2006.8 | 1909.3 | −4.9 |
| South Africa | 6624.8 | 5655.9 | −14.6 |
| Tunisia | 3691 | 3597 | −2.6 |
| Seychelles | 17,252 | 13,306.7 | −22.9 |

*3.3. Death Rate and GDP before and after the Outbreak*

Figure 4 displays the GDP of African countries. The figure shows that Nigeria's GDP is the largest (504 billion dollars), while Sao Tome and Principe's is the smallest (0.5 billion). The average GDP for 54 African countries is 62.78 billion USD.

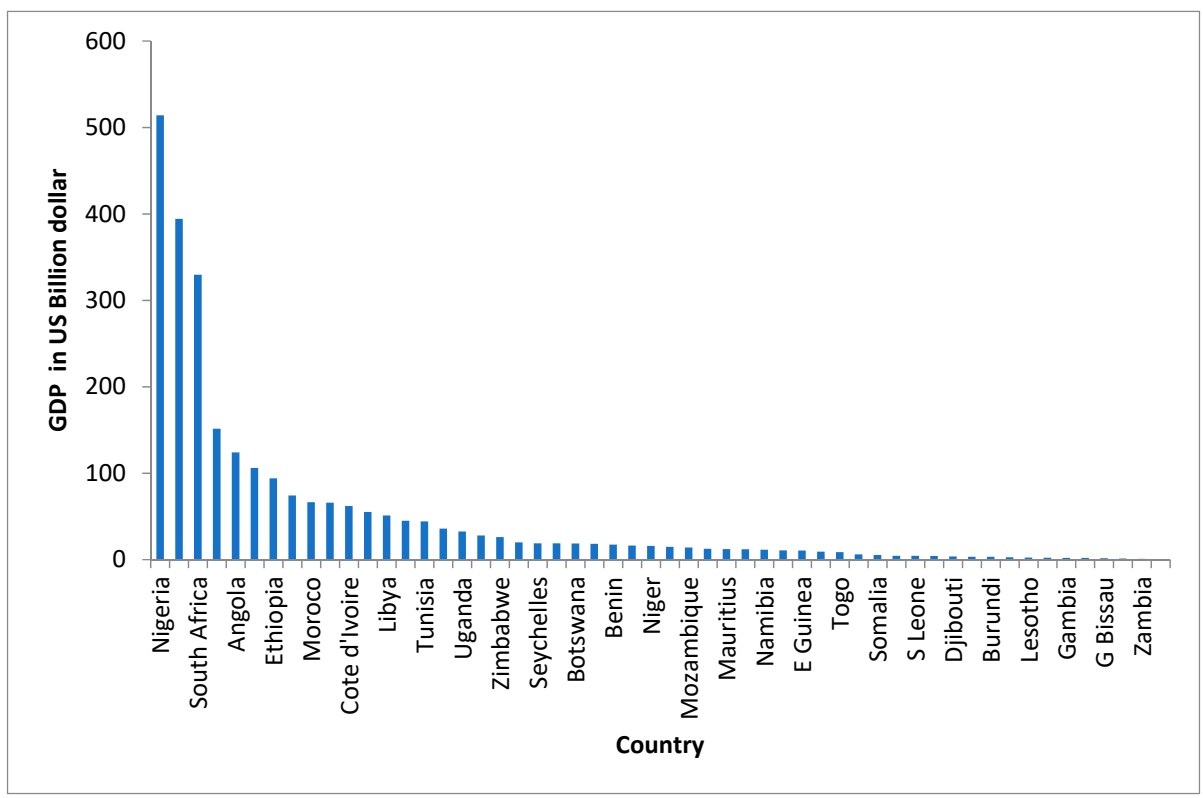

**Figure 4.** GDP in billions of USD by country in Africa.

Figure 5 depicts the number of COVID-19 cases per 1000 people by country. It shows that South Africa experienced the highest number of cases (3648.968 per 1000 people), followed by Tunisia (974.214 per 1000 people).

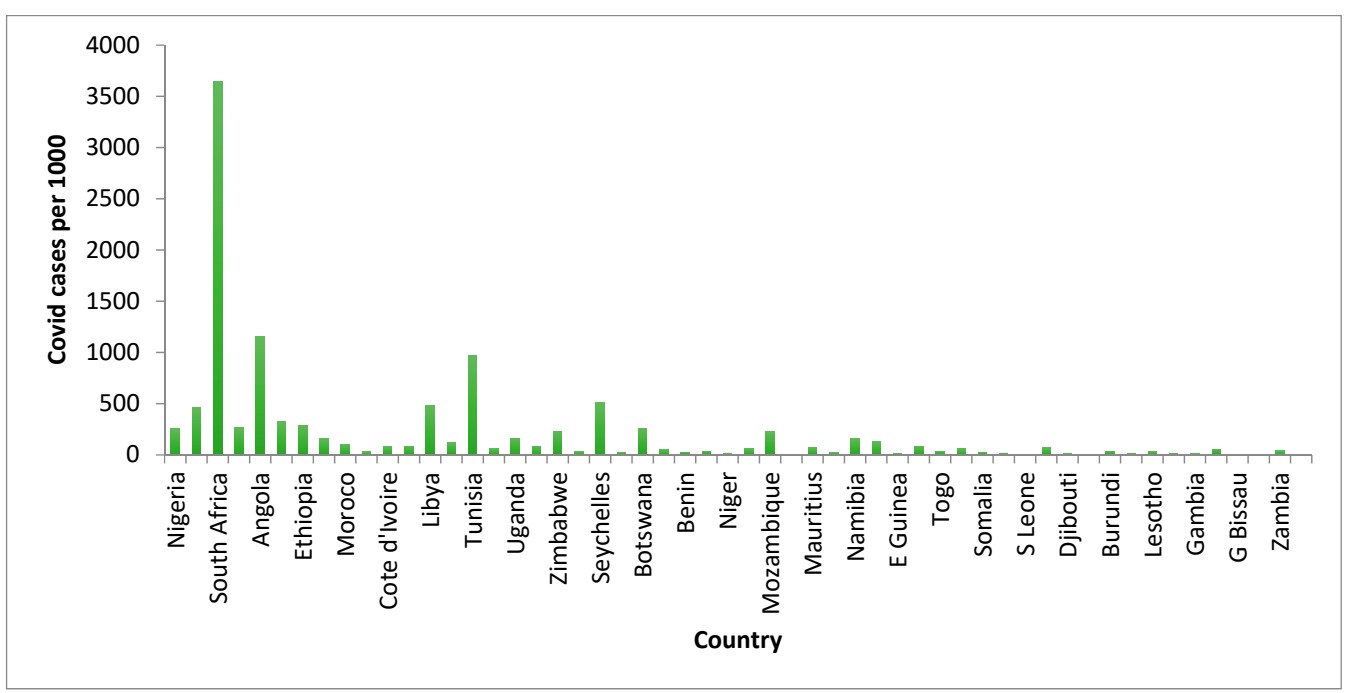

**Figure 5.** COVID-19 cases per 1000 people in Africa.

Figure 6 depicts the death rates due to COVID-19 for selected countries, from which we can clearly observe the variation in the spatial distribution of the death rates. The result shows that South Africa, Angola, Tunisia, and Ethiopia have experienced high death rates due to the pandemic in Africa.

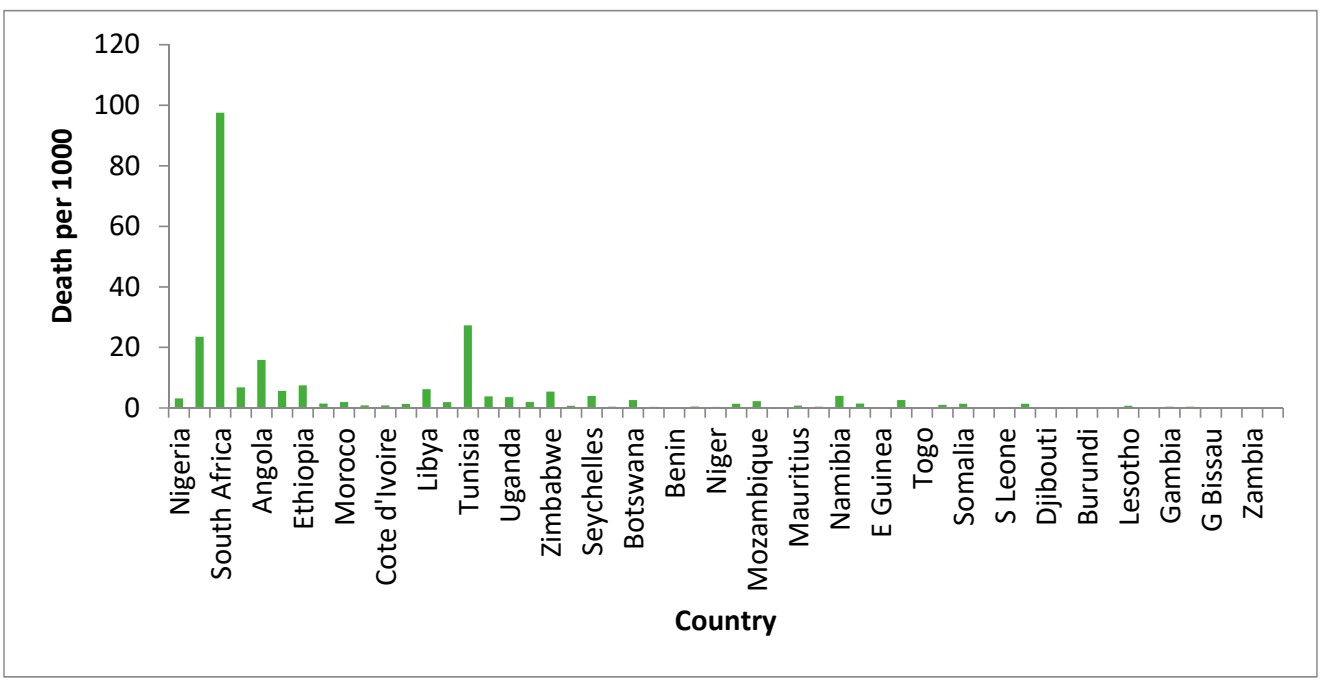

**Figure 6.** The spatial distribution of deaths due to COVID-19 per 1000 people.

*3.4. Testing for Spatial Autocorrelation*

The Moran's I coefficient, one of the most extensively used measures of spatial autocorrelation, was utilized to identify the spatial patterns. Tests and presentations of the global Moran's I and local Moran's I statistics for clustering are included in the spatial autocorrelation analysis package. The Moran scatter plot is used to show the global test, in which the slope of the regression line corresponds to Moran's I. At a significance level of 0.05, the non-existence of significant clustering of the death rate and confirmed cases due to COVID-19 was tested in selected countries. First, we calculated the global *Moran's I test statistics*. Furthermore, to ensure that the results were consistent, a diagnostic test for spatial dependence was conducted.

3.4.1. Tests of Spatial Autocorrelation Using Global Moran's I

The spatial distribution of COVID-19 cases and the death rate in Africa can be critically examined using the global spatial autocorrelation. Our goal here is to test the null hypothesis (under the normality and independence assumptions) of "there is no spatial autocorrelation ($\rho = 0$)" against the alternative hypothesis of "spatial dependence/autocorrelation ($\rho \neq 0$)". The result of global Moran's I shows that there is a positive spatial autocorrelation for the confirmed cases of COVID-19 (Moran's I = 0.3432, *p*-Value = 0.0100) and death rate (Moran's I = 0.3895, *p*-Value = 0.0100) at the 5% level of significance. The result corresponding to the global test of spatial autocorrelation is given in Table 4.

**Table 4.** The results of the global Moran's I correlation coefficient.

| Variable | Moran I Correlation under Normalization | | | | | |
|---|---|---|---|---|---|---|
| | Coefficient | Observed | Expected | Std | Z Value | *p*-Value |
| Death per 1000 people | Moran's I statistic | 0.3895 | −0.0204 | −0.0397 | 6.088 | 0.01 |
| COVID-19 cases per 1000 people | Moran's I statistic | 0.3432 | −0.0204 | 0.0993 | 3.6373 | 0.01 |

Based on the *p*-values of the provided Moran's I coefficients, the global tests of spatial autocorrelation suggest the rejection of the null hypothesis of no spatial autocorrelation of the two spatial components at the 5% level of significance. The Moran's I coefficients of the confirmed cases and death rate show that the death rate due to COVID-19 and confirmed cases have substantial positive spatial autocorrelation or clustering. Under the assumption of normality, we utilize Moran's scatter plot to depict global spatial autocorrelation, as shown in Figure 7a,b.

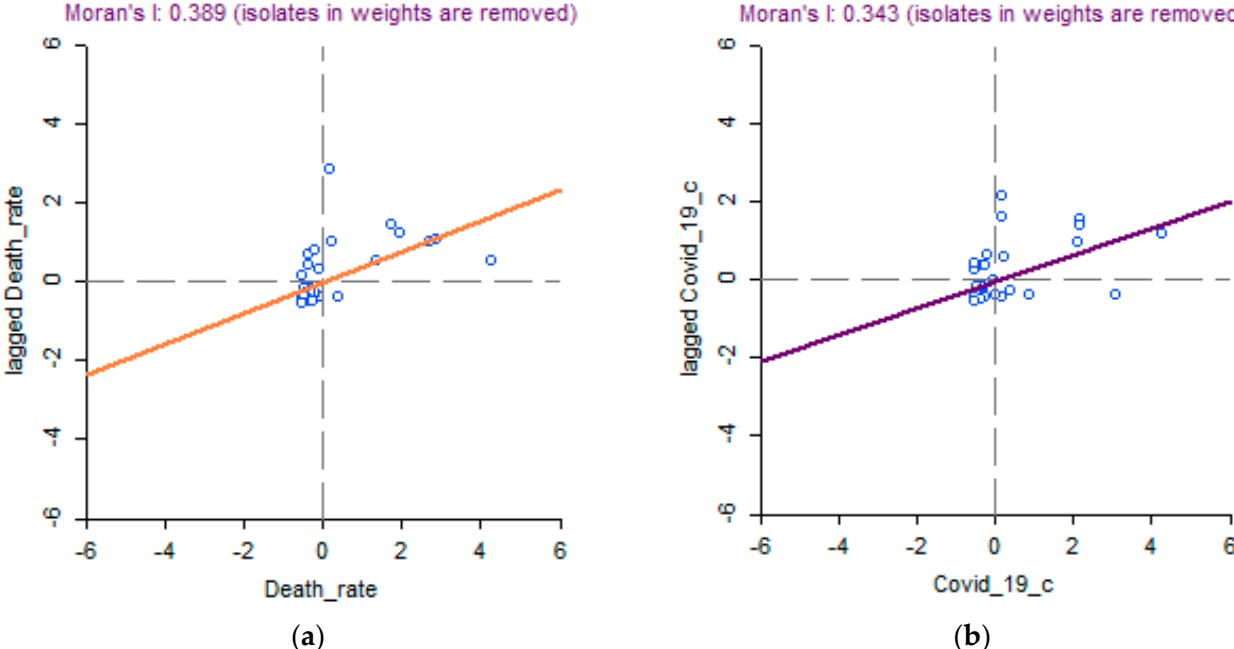

(**a**)                                                                      (**b**)

**Figure 7.** The Moran scatter plot of the spatial distribution of confirmed cases and death rate. The result shown in both (**a**,**b**) show that there is spatial dependence two locations based on the data of death rate and number of confirmed cases of COVID 19. The values of the Moran's I reveals that there is positive spatial autocorrelation confirming the existance of the spatial autocorrelation on both events.

Figure 8a presents the spatial distribution of the percentage decrease in GDP per capita by comparing the values of GDP before and after the pandemic in African countries. It can be observed that in countries where there are fewer COVID-19 cases, the percentage decrease in GDP per capita in USD is low. On the contrary, in countries where there are high COVID-19 cases, the percentage decrease in GDP per capita in USD is high. Our results are similar to those of the study conducted by [38]. Following the result corresponding to the spatial distribution of GDP in Africa, the geospatial result corresponding to the COVID-19 cases and death rate in Africa is visualized in Figure 8b,c, from which we note that South Africa was the most highly affected country compared to the other African countries.

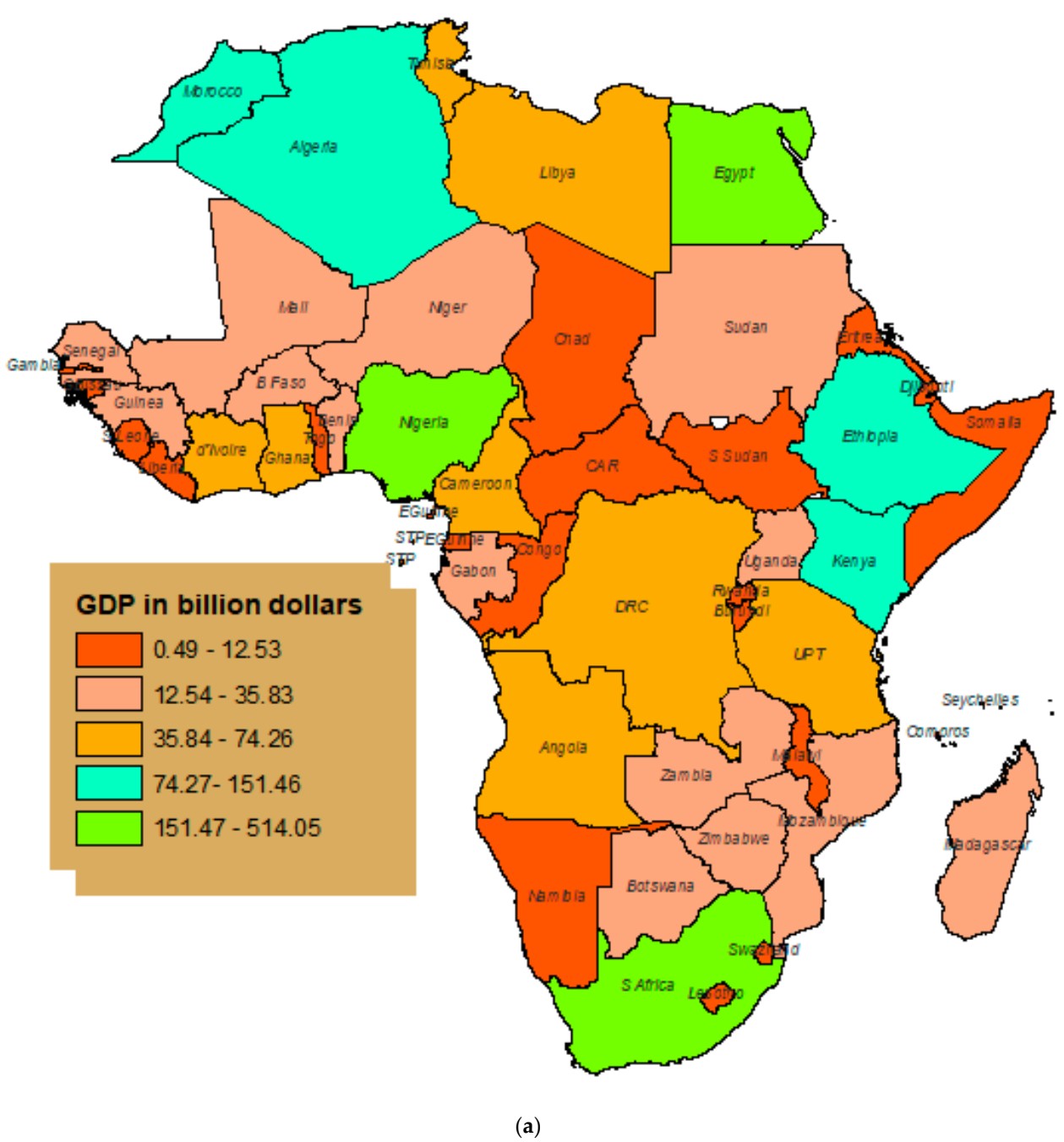

(**a**)

**Figure 8.** *Cont.*

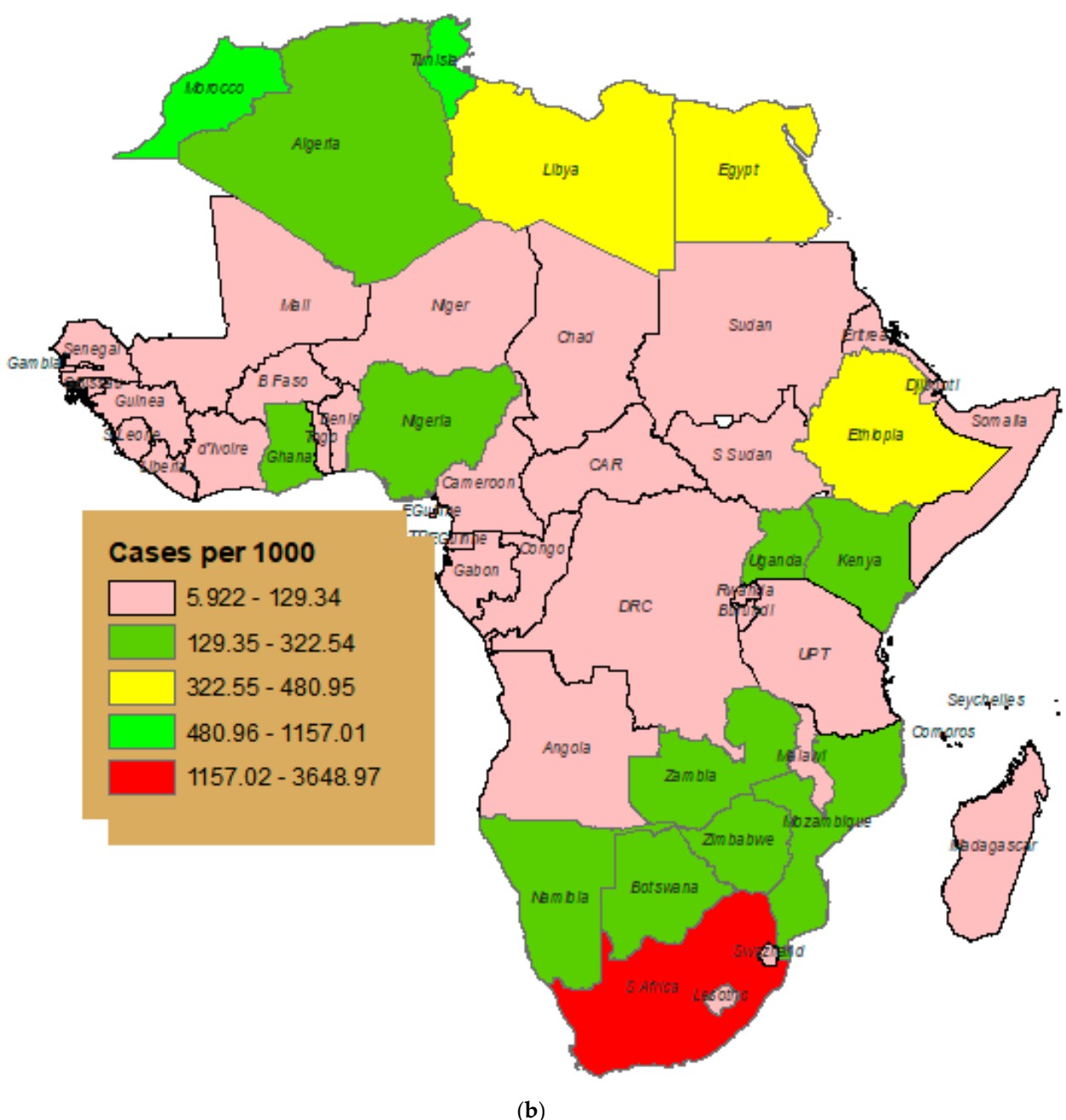

(**b**)

**Figure 8.** *Cont.*

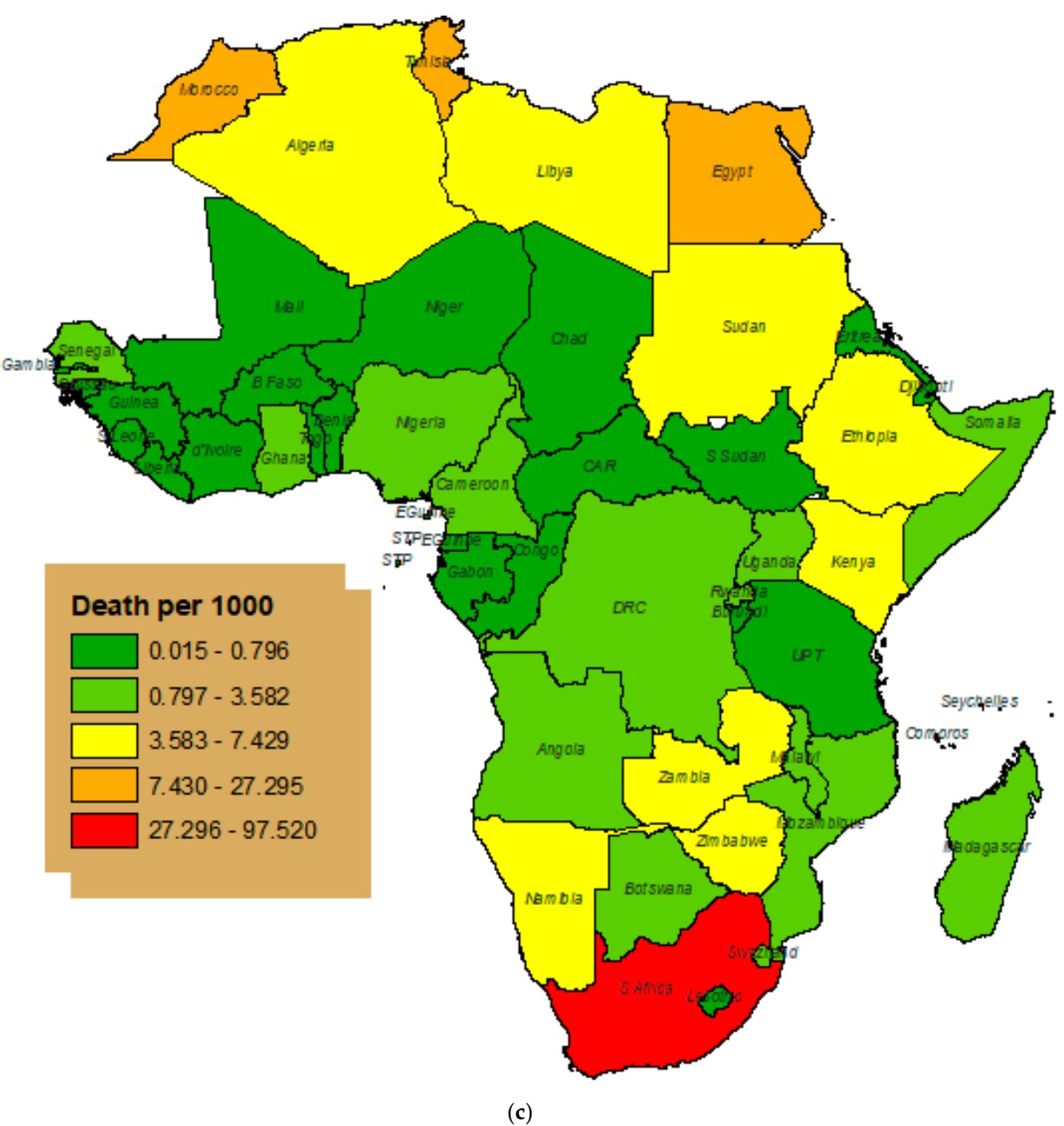

(**c**)

**Figure 8.** GDP per capita distribution (**a**), hotspot areas of confirmed cases (**b**), and deaths due to COVID-19 per 1000 people (**c**).

3.4.2. Result of Multivariate Analysis of Covariance Methods

To determine the relationship between several predictor variables and the confirmed cases and death rate due to the pandemic, we also conducted a multivariate analysis of covariance and obtained the results presented in Table 5.

The result of multivariate analysis of covariance shows that death rate due to COVID-19 with population density, temperature with COVID-19 cases and death rate, precipitation with death rate due to COVID-19, and wind with death rate due to COVID-19 have a significant association at the 5% level of significance (see Table 5).

**Table 5.** Effect of predictor variables on COVID-19 in Africa.

| Source | Dependent Variable | Type III Sum of Squares | DF | Mean Square | F | Sig. |
|---|---|---|---|---|---|---|
| | | **Tests Between-Subjects Effects** | | | | |
| Corrected | Confirmed | 725,709,352.30 | 5 | 145,141,870.50 | 7.514 | 0.000 |
| Model | Death | 88,008.786 | 5 | 17,601.757 | 21.241 | 0.000 |
| Intercept | Confirmed | 143,780,213.00 | 1 | 143,780,213.00 | 7.443 | 0.009 |
| | Death | 15,338.553 | 1 | 15,338.553 | 18.51 | 0.000 |
| Population density | Confirmed | 36,485,798.84 | 1 | 36,485,798.840 | 1.889 | 0.175 |
| | Death | 8391.913 | 1 | 8391.913 | 10.127 | 0.002 |
| Temperature | Confirmed | 258,922,007.10 | 1 | 258,922,007.10 | 13.40 | 0.001 |
| | Death | 19,730.240 | 1 | 19,730.240 | 23.810 | 0.000 |
| Precipitation | Confirmed | 40,568,504.63 | 1 | 40,568,504.63 | 2.100 | 0.153 |
| | Death | 3227.607 | 1 | 3227.607 | 3.895 | 0.054 |
| Humidity | Confirmed | 77,339,061.270 | 1 | 77,339,061.270 | 4.004 | 0.051 |
| | Death | 1544.560 | 1 | 1544.560 | 1.864 | 0.178 |
| Wind | Confirmed | 7,421,844.831 | 1 | 7,421,844.831 | 0.384 | 0.538 |
| | Death | 4768.626 | 1 | 4768.626 | 5.755 | 0.020 |
| Error | Confirmed | 102,381,7543.0 | 47 | 19,317,312.140 | | |
| | Death | 43,918.572 | 47 | 828.652 | | |
| Total | Confirmed | 198,669,3983 | 54 | | | |
| | Death | 174,588.504 | 54 | | | |

### 3.4.3. Inflation and Unemployment Rate (before and during COVID-19)

This subsection addresses the level of inflation before and during the outbreak of COVID-19 in Africa. The study also explored the effect of COVID-19 on the unemployment rate based on the data taken from the employment and total labor force reports from the World Bank. Figure 9 shows the inflation after the outbreak of the pandemic as compared with the inflation before the outbreak of the pandemic in each country, from which we note that there has been a significant increment in inflation since the outbreak of the pandemic. We considered the consumer price index before the outbreak of the pandemic and the consumer price index of 2020 and 2021 during the pandemic for each country. As depicted in Figure 9, the inflation after the outbreak of COVID-19 for each country is high as compared to the inflation before the outbreak of the pandemic. The result reveals that there is a marked difference in inflation before and after the outbreak of the pandemic in all countries (Figure 9).

Table 6 shows the inflation before and after the outbreak of COVID-19 in Africa. The result clearly shows a marked difference in inflation before and after the outbreak of the pandemic in Africa.

**Table 6.** Inflation before and during COVID-19 in Africa.

| Country | Inflation before COVID-19 | Inflation after COVID-19 |
|---|---|---|
| Angola | 18.35 | 23.85 |
| Benin | −0.05 | 5.92 |
| Burkina Faso | −0.64 | 3.85 |
| Botswana | 3.03 | 7.24 |
| Central African | 2.15 | 3.34 |
| Côte d'Ivoire | 0.62 | 4.09 |
| Cameroon | 1.76 | 2.27 |

**Table 6.** *Cont.*

| Country | Inflation before COVID-19 | Inflation after COVID-19 |
|---|---|---|
| Congo | 16.99 | 21.89 |
| Congo, Rep. | 1.68 | 1.97 |
| Djibouti | 1.73 | 2.35 |
| Algeria | 2.73 | 7.23 |
| Egypt | 14.14 | 5.21 |
| Ethiopia | 14.83 | 26.84 |
| Gabon | 3.40 | 5.13 |
| Ghana | 8.51 | 9.97 |
| Guinea | 9.65 | 12.60 |
| Gambia | 6.82 | 7.37 |
| Guinea-Bissau | 0.84 | 3.25 |
| Equatorial Guinea | 1.15 | 12.10 |
| Kenya | 4.95 | 6.11 |
| Liberia | 25.26 | 30.86 |
| Libya | 1.68 | 18.24 |
| Sri Lanka | 3.22 | 7.01 |
| Lesotho | 4.60 | 6.05 |
| Madagascar | 6.46 | 5.40 |
| Mozambique | 3.35 | 5.69 |
| Malawi | 9.30 | 9.47 |
| Namibia | 4.00 | 3.62 |
| Niger | 0.25 | 3.84 |
| Nigeria | 11.75 | 16.95 |
| Rwanda | 1.89 | 10.39 |
| Sudan | 57.14 | 59.09 |
| Senegal | 0.74 | 2.18 |
| Chad | 1.65 | 10.77 |
| Togo | 0.81 | 4.55 |
| Tonga | 3.10 | 10.15 |
| Tunisia | 7.00 | 15.71 |
| Tanzania | 3.48 | 3.69 |
| Uganda | 2.74 | 12.21 |
| South Africa | 4.32 | 4.61 |
| Zambia | 8.65 | 22.02 |
| Zimbabwe | 32.95 | 98.55 |

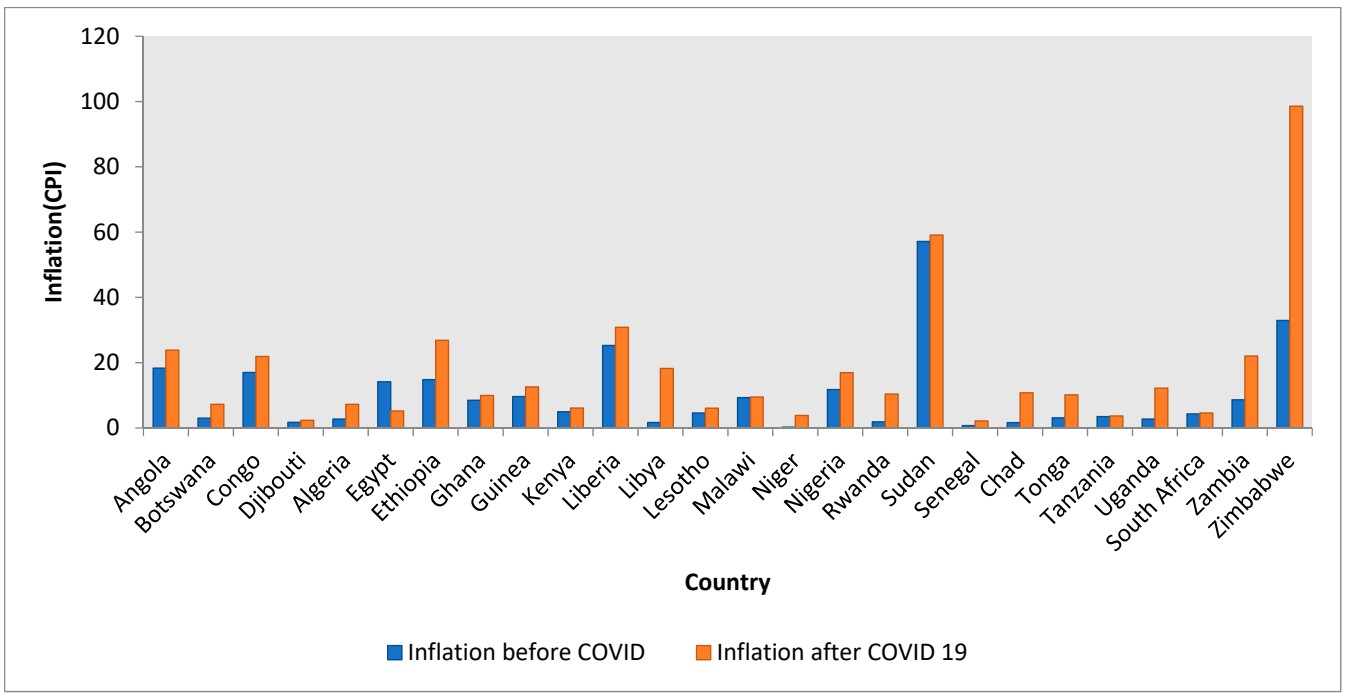

**Figure 9.** Inflation before and during COVID-19.

Table 7 confirms the existence of a significant mean difference in inflation before and during the outbreak of the pandemic in Africa.

**Table 7.** Paired sample test: Inflation before and during COVID-19 in Africa.

| | Paired Samples Test | | | | | | | |
| --- | --- | --- | --- | --- | --- | --- | --- | --- |
| | Paired Differences | | | | | | | |
| | Mean | Std. Deviation | Std. Error Mean | 95% Confidence Interval of the Difference | | t | df | Sig. (2-Tailed) |
| | | | | Lower | Upper | | | |
| Inflation before and during the outbreak of COVID-19 | −5.39643 | 10.52967 | 1.62476 | −8.67770 | −2.11515 | −3.321 | 41 | 0.002 |

The result of the study also showed that inflation during COVID-19 was extremely high and has indirectly influenced the GDP per capita in African countries. Similarly, the unemployment rate during the pandemic has also increased considerably in African countries, ranging from a minimum of 0.6875 in Niger to a maximum of 31.3895 in South Africa (Figure 10).

### 3.5. Result of Spatial Autoregressive Modeling

The analysis using a spatial autoregressive model focused on the relationship between confirmed cases, death per 1000 people due to COVID-19, and inflation on the one hand and GDP per capita on the other. The result shows that as inflation increases, the GDP per capita decreases consistently across African countries. Similarly, as confirmed cases and deaths due to COVID-19 increase, GDP per capita declines across the countries.

Several researchers have studied the benefits of environmental, social, and governance (ESG) performance. For instance, Ref. [39] has assessed the importance of good ESG performance for GDP per capita and suggested the existence of a positive relationship between ESG and GDP in terms of per capita over the long term. Therefore, a decrease in GDP per capita is somehow related to inefficient ESG performance. This indicates that

in order to bring about sustainable economic recovery in the post-COVID-19 era, there is a need for coordinated efforts to improve the performance of ESG in African countries, focusing on areas, such as investment in renewable energy, promoting sustainable business practices, and supporting local communities.

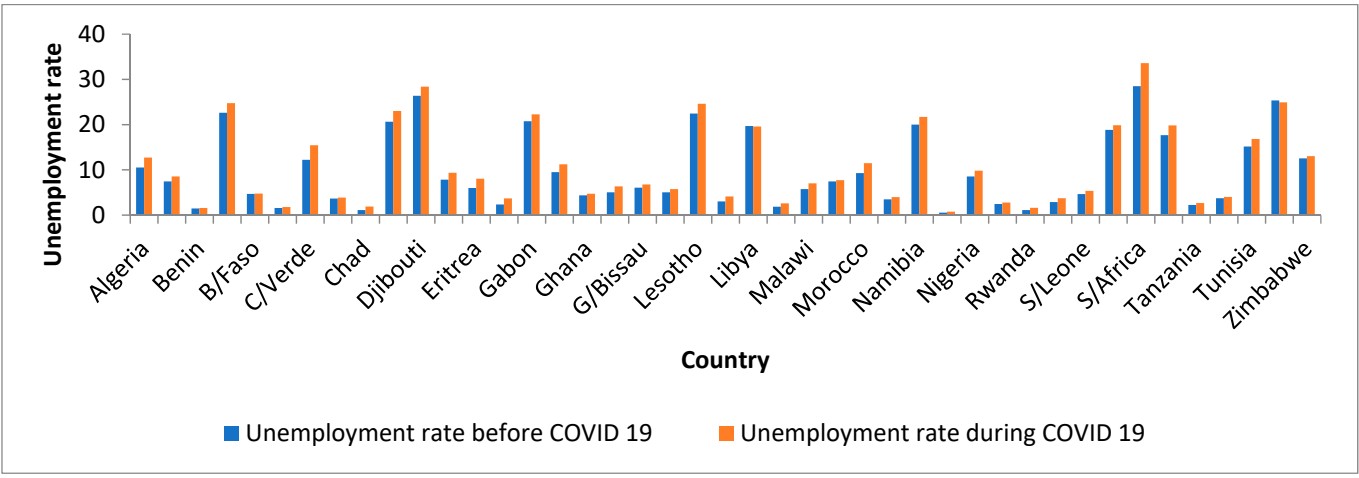

**Figure 10.** Unemployment rate before and during the pandemic.

## 4. Conclusions

In this research article, we assessed the effects of COVID-19 on inflation, unemployment rate, and GDP per capita in Africa via GIS and spatial statistics. Measures of spatial autocorrelation, including Moran's I and Moran scatter plots, were employed to visualize the spatial distribution of the three parameters (GDP, death per 1000 people due to COVID-19, and COVID-19 cases per 1000 people). The resuts indicate that the spatial distribution of the pandemic is clustered, justifying the existence of spatial autocorrelation. The result of the study also shows a decrease in GDP per capita during the pandemic, and the rate of decrease varies from country to country, confirming the existence of spatial dependency. A significantly high death rate and a high confirmed number of cases of COVID-19 were related to a low GDP per capita. The result of multivariate analysis of covariance shows that there is a significant association between death rate due to COVID-19 and population density, temperature with COVID-19 cases, precipitation with death rate due to COVID-19, and wind with death rate due to COVID-19 at a 5% level of significance. There is also evidence of a significant difference between the consumer price index (inflation rate) and unemployment rate registered before and after the outbreak of the pandemic. A marked increase in inflation and unemployment rates has been observed since the outbreak of COVID-19. Reducing unemployment and boosting the GDP per capita can boost the performance of the ESG. The study suggests the need to support African countries in reducing/controlling the impact of COVID-19 on inflation, unemployment, and GDP per capita.

**Author Contributions:** The corresponding author (H.T.) designed the manuscript, wrote the whole manuscript, and conducted the analysis. The author, B.G., edited and revised the entire work. All authors have read and agreed to the published version of the manuscript.

**Funding:** This research received no external funding.

**Institutional Review Board Statement:** We declare that this work is very ethical and has no problems related to the issue of ethics. The authors are authorized to download data from the IMF and WHO. The data is publicly available and has no personal identifiers.

**Informed Consent Statement:** Not applicable.

**Data Availability Statement:** The data presented in this study are available on request from the corresponding author.

**Acknowledgments:** The authors would like to thank the IMF and WHO for providing the data for the study.

**Conflicts of Interest:** The authors declare no conflict of interest.

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
