# Peer review of "Assessing COVID-19 Effects on Inflation, Unemployment, and GDP in Africa: What Do the Data Show via GIS and Spatial Statistics?"

_covid, doi:10.3390/covid3070069_

Round 1

Reviewer 1 Report

Dear COVID Editorial Office,        

Dear Authors, Thank You for the relevant article entitled "Assessing COVID-19 effects on inflation, unemployment and Gross Domestic Product in Africa: What Do the data show via GIS and Spatial Exploratory Statistics?". I'd like make some recommendations. 1. The research aim of this paper is to assess the effect of COVID-19, inflation, and unemployment rate on Gross Domestic Product (GDP) in Africa via GIS and Spatial Exploratory Analysis. The study also aims to map the spatial distribution of COVID-19 in Africa and examine its impact on the economy. The authors have obtained data from the IMF and WHO and have conducted a thorough analysis. 2. The relevance of the study is that the COVID-19 pandemic has had a significant impact on the global economy, and Africa is not an exception. The research gap is that there is a lack of comprehensive studies that assess the impact of COVID-19, inflation, and unemployment rate on the GDP per capita in Africa using GIS and spatial exploratory analysis. This study aims to fill this gap by providing a comprehensive analysis of the impact of these factors on the African economy. 3. The study employed a cross-sectional study design along with spatial exploratory analysis and multivariate analysis of covariance (MANCOVA). The entire dataset was analyzed with the help of Stata, ArcGIS 10.4.1, and GeoDa software. The spatial distribution of COVID-19 and GDP per capita was mapped using ArcGIS, while spatial autocorrelation, including the Moran's I and Moran scatter plot, was employed to visualize the spatial distribution of the variables. Multivariate analysis of covariance (MANCOVA) was used to examine the relationship between the variables. 4. The study concludes that the COVID-19 pandemic has had a significant impact on the African economy, particularly on the GDP per capita. The study found that there is a significant association between death rate due to COVID-19 and population density, temperature with COVID-19 cases and death rate, precipitation with death rate due to COVID-19, and wind with death rate due to COVID-19. The study also found that significant high death and confirmed number of cases of COVID-19 were associated with low GDP per capita. The assessment indicates a significant difference between consumer price index (inflation rate) and unemployment rate registered before and after the outbreak of the pandemic, where a marked increment of inflation and unemployment rate were observed since the outbreak of COVID-19. The study suggests the need for supporting African countries in reducing/controlling the impact of COVID on inflation, unemployment, and GDP per capita.  5. The references used are credible and relevant to the research topic. The research cites a range of relevant and credible sources (35 sources are in the list of the references), including reports from the World Health Organization, data from the International Monetary Fund, and research articles published in peer-reviewed journals. The references used in the study appear to be sufficient to support the study's findings and conclusions.  7. The study highlights the impact of COVID-19 on GDP per capita, inflation, and unemployment rate in Africa. Policymakers can use this information to develop policies that promote sustainable economic recovery, such as investing in renewable energy, promoting sustainable business practices, and supporting local communities. I'd suggest the Authors should consider the Sustainability concept in the discussion section of the Article. By incorporating some discussion on the social, environmental and governing dimensions of ESG-goal achievement, the authors can contribute to a more holistic understanding of the role of the sustainable development in post-pandemic economy. 8. I'd recommend the Authors should disclose the sentence in lines 229-230: "It can be seen that the GDP per capita has decreased during the pandemic as compared to GDP per capita before the outbreak of the pandemic for all countries except Egypt." I could not find in the text any explanation for this issue. How do the Authors understand this exception? Such explanation would improve the whole idea. 9. Overall, I find this article to be an exciting and relevant contribution to the field of the post-pandemic socio-economic development, and I recommend it for publication in the MDPI Journal "COVID" after minor revision.

Quality of English is sufficient high and requires only minor editing

Author Response

The authors would like to express their gratitude to the anonymous reviewers for carefully reviewing the manuscript and for many thoughtful comments, which have enhanced the readability and quality of this manuscript. For easy check-up, our responses are given in color.

Reviewer#1

Overall, I find this article to be an exciting and relevant contribution to the field of the post-pandemic socio-economic development, and I recommend it for publication in the MDPI Journal "COVID" after minor revision.

Reviewer#1, Concern # 1 

Policymakers can use this information to develop policies that promote sustainable economic recovery, such as investing in renewable energy, promoting sustainable business practices, and supporting local communities. I'd suggest the Authors should consider the Sustainability concept in the discussion section of the Article. By incorporating some discussion on the social, environmental and governing dimensions of ESG-goal achievement, the authors can contribute to a more holistic understanding of the role of the sustainable development in post-pandemic economy. 

Author response:  

Point is well taken. The discussion corresponding to concern one is briefly included in the final version of the manuscript including  conclusion subsection (see the updates marked in color).

Reviewer#1, Concern # 2

I'd recommend the Authors should disclose the sentence in lines 229-230: "It can be seen that the GDP per capita has decreased during the pandemic as compared to GDP per capita before the outbreak of the pandemic for all countries except Egypt." I could not find in the text any explanation for this issue. How do the Authors understand this exception? Such explanation would improve the whole idea.

Author’s response: 

Point is well taken.  The justification why the economy during the COVID-19 is not impacted is mentioned in the latest version of the manuscript. The possible reason why the GDP of Egypt is relatively not influenced much due to the pandemic is described and updated in the latest manuscript. 

Reviewer 2 Report

- The abstract is extremely long. Please bring that to half size.

- Line 18: Assessing, Inflation

- Line 33: decrease of 14.6

- Please do not use unexplained abbreviations in the abstract - nor in the text afterwards.

- The citation on the left side of page 1 does not entail the entire title of the paper.

- Line 73: please avoid stating words like "some". Please be precise.

- Line 86-87: cursive

- Figure 1: Cape Verde seems to appears twice. However, the figure is not really readable.

- Line 114: please do not indicate solely URL's - they change often very fast

- Line 134: besides naming the reference, please also bring in a reference with number

- Line 249: 974.214

- Figure 8 is not readable

- Figure 13 has a very large space at the right side

- Please elaborate the conclusions - much more could be said here.

English is ok

Author Response

The authors would like to express their gratitude to the anonymous reviewers for carefully reviewing the manuscript and for many thoughtful comments, which have enhanced the readability and quality of this manuscript. For easy check-up, our responses are given in color.

Reviewer#2, Concern # 1 

The abstract is too long. Make it short

Author’s response: 

Point is well taken.  We update the abstract by reducing the size considering only the main important points. Thus, the abstract is modified.

Reviewer#2, Concern # 2:

Line 18: assessing, inflation

Author’s response: 

Point is well taken. The minor typographical errors are corrected and the latest manuscript is updated accordingly (see the latest version).

Reviewer#2, Concern # 3:

Line 33: decrease 14.6

Author’s response: 

Point is well taken taken.  It is a typo error. Thus, all typos across the entire manuscript are corrected once for all. See again the latest version of the article. 

Reviewer#2, Concern # 4:

Please don’t use unexplained abbreviations in the abstract and afterwards

Author’s response: 

Point is well taken: We have revised the abstract and the entire document explaining the abbreviations and updated the manuscript accordingly.

Reviewer#2, Concern # 5:

Please avoid the words like some in the article

Author’s response: 

Point is well taken and manuscript is updated.

Reviewer#2, Concern # 6:

Line 114 please don’t indicate solely URL’s they change often fast. 

Author’s response: 

Point is well taken: We updated manuscript removing URL’s and citing in its place properly.

Reviewer#2, Concern # 7:

Line 249: 974.214. 

Author’s response: 

Point is well taken: yes the font size of the number is corrected and the document is also updated.

Reviewer#2, Concern # 8:

Figure 8 is not readable

Author’s response: 

Point is well taken: Figure 8 modified.

Reviewer#2, Concern # 9:

Figure 13 has a very large space at the right side. 

Author’s response: 

Fig 12 and Fig 13 are removed and a brief text is added in the latest version to address the point.

Reviewer#2, Concern # 11:

Please elaborate the conclusion much more could be said here. 

Author’s response: 

The conclusions are drawn based on the research objectives and findings. As such, we also revised a bit adding some more into the latest version of the manuscript. See the updated manuscript.

Reviewer#2, Concern # 11:

Minor editing of English language required

Author’s response: 

Point is well taken. We have made line editing.

Round 2

Reviewer 2 Report

Review suggestions have been well implemented.